# The Crystal Structure of Tyrosinase from *Verrucomicrobium spinosum* Reveals It to Be an Atypical Bacterial Tyrosinase

**DOI:** 10.3390/biom13091360

**Published:** 2023-09-07

**Authors:** Mostafa Fekry, Khyati K. Dave, Dilip Badgujar, Emil Hamnevik, Oskar Aurelius, Doreen Dobritzsch, U. Helena Danielson

**Affiliations:** 1Department of Chemistry—BMC, Uppsala University, SE 751 23 Uppsala, Sweden; mostafa.fekry.abdalkhalik@kemi.uu.se (M.F.); khyatikdave@gmail.com (K.K.D.); dilip.badgujar@icm.uu.se (D.B.); emil.hamnevik@gmail.com (E.H.); doreen.dobritzsch@kemi.uu.se (D.D.); 2Biophysics Department, Faculty of Science, Cairo University, Giza 12613, Egypt; 3MAX IV Laboratory, Lund University, 22100 Lund, Sweden; oskar.aurelius@maxiv.lu.se; 4Science for Life Laboratory, Drug Discovery & Development Platform, Uppsala University, SE 751 23 Uppsala, Sweden

**Keywords:** tyrosinase, *Verrucomicrobium spinosum*, crystal structure

## Abstract

Tyrosinases belong to the type-III copper enzyme family, which is involved in melanin production in a wide range of organisms. Despite similar overall characteristics and functions, their structures, activities, substrate specificities and regulation vary. The tyrosinase from the bacterium *Verrucomicrobium spinosum* (*vs*Tyr) is produced as a pre-pro-enzyme in which a C-terminal extension serves as an inactivation domain. It does not require a caddie protein for copper ion incorporation, which makes it similar to eukaryotic tyrosinases. To gain an understanding of the catalytic machinery and regulation of *vs*Tyr activity, we determined the structure of the catalytically active “core domain” of *vs*Tyr by X-ray crystallography. The analysis showed that *vs*Tyr is an atypical bacterial tyrosinase not only because it is independent of a caddie protein but also because it shows the highest structural (and sequence) similarity to plant-derived members of the type-III copper enzyme family and is more closely related to fungal tyrosinases regarding active site features. By modelling the structure of the pre-pro-enzyme using AlphaFold, we observed that Phe453, located in the C-terminal extension, is appropriately positioned to function as a “gatekeeper” residue. Our findings raise questions concerning the evolutionary origin of *vs*Tyr.

## 1. Introduction

Tyrosinases (EC.1.14.18.1) are bifunctional metalloenzymes catalyzing two sequential enzymatic reactions: hydroxylation of monophenols to o-diphenols (monophenolase activity) and oxidation of o-diphenols to the corresponding o-quinones (diphenolase or catecholase activity) [1,2,3]. The polymerization of reactive quinones results in melanin, a structurally and functionally diverse class of compounds involved in skin pigmentation, the browning of vegetables and fruits, and many other biological processes [4,5,6]. Tyrosinases (Tyr) are widely distributed in mammals, plants, fungi, and bacteria. They belong to the group of type-III copper enzymes known as polyphenol oxidases (PPOs). This group also includes catechol oxidase (CO) and its plant counterpart, aurone synthase (AUS). COs have diphenolase activity but lack monooxygenase activity [7,8]. Laccases and hemocyanins (Hc) are also related to PPOs. The latter primarily function as oxygen carriers within the hemolymph of mollusks and arthropods, but also have enzymatic activity and additional functions [9].

Members of the type-III copper protein family contain a characteristic binuclear copper center, in which each copper ion (CuA and CuB) is coordinated by three highly conserved histidine residues. Additional residues identified as being crucial for catalysis and substrate specificity are a “gatekeeper” amino acid (or blocker residue) located near CuA, and a “waterkeeper” residue. The gatekeeper is supposed to influence copper ion incorporation, to stabilize the substrate in the active site, and to control enzymatic activity in order to prevent undesirable oxidation of phenolic compounds. The waterkeeper, positioned at the entrance of the active site, is thought to be involved in the deprotonation of monophenolic substrates [10].

Although the copper centers and core structures of the proteins within this family are highly conserved, their potential enzymatic activities are different. This is a consequence of both the accessibility of their active site for a variety of substrates and of how the substrates are stabilized and activated in the active site [11,12,13,14,15,16,17]. In two recent studies, site-directed mutagenesis was used to convert the tyrosinase from walnut (*Juglans regia*, *jr*Tyr) into a catechol oxidase [18] and to establish the role of residues around the CuA site on the catalytic properties of *Coreopsis grandiflora* aurone synthase (*cg*AUS) [19]. The substitutions involved the gatekeeper and additional activity controller residues, which could thus be identified as determinants of the absence or presence of monophenolase activity in the diverse members of the protein family.

Tyrosinases and other PPOs all contain a conserved catalytic or core domain, but based on the presence of additional domains and the mechanism of enzyme activation, these enzymes can be divided into different groups [14,20,21,22,23,24,25]. Eukaryotic tyrosinases feature a C-terminal extension that regulates enzyme activity by blocking substrate entrance to the active site and that may also play a role in copper incorporation [26]. In contrast, most bacterial tyrosinases (e.g., *Streptomyces *sp.) require an accessory metallochaperone, a “caddie protein”, for correct folding, copper ion incorporation and activity [3]. The tyrosinase from the bacterium* Verrucomicrobium spinosum* (*vs*Tyr) is an exception in this group as it shows low sequence identity (<30% overall, i.e., when comparing full-length sequences) to other bacterial tyrosinases and does not rely on a caddie protein for copper ion incorporation but contains a C-terminal extension typical for plant and fungal enzymes.

The full-length *vs*Tyr consists of 482 amino acids comprising three domains in total (Figure 1). The N-terminal domain (1–36) encompasses a twin-arginine translocase (TAT) signal peptide responsible for the export of the protein to the periplasmic space in folded form. The core domain (37–357) harbors the active site with the binuclear copper center. The C-terminal domain (358–518) keeps the enzyme inactivated until needed, whereupon it is likely removed by proteolytic cleavage, resulting in an increase in active site accessibility and enzymatic activity [27]. The recombinant core domain of *vs*Tyr, expressed without the C-terminal extension, has previously been shown to have a very high specific activity and to be unusually resistant to denaturing agents [28]. Owing to these properties, *vs*Tyr may be better suited for different biotechnological applications than the *Agaricus bisporus* tyrosinase (*ab*Tyr) used to date [29].

In this study, we report on the crystal structure of the *vs*Tyr core domain and identify structural features of importance for substrate specificity and the regulation of its enzymatic activity.

## 2. Materials and Methods

### 2.1. Cloning, Expression and Purification of the Pro-Tyrosinase and Core Domain of vsTyr

The DNA coding sequence of pro-*vs*Tyr (amino acids 36–518, NCBI Reference Sequence WP_009958178.1) was synthesized by Invitrogen GeneArt services (Thermo Fisher Scientific, Waltham, MA, USA) and subsequently cloned into the *SspI–KpnI* site of a pETHis–LIC cloning vector (Addgene, Watertown, MA, USA) to obtain pETHis–protyr. See Appendix A for the complete amino acid sequence and Figure 1 for an overview of the domain structure with amino acid numbers specifying the different domains (the numbering used is as in reference [27]). The core domain of *vs*Tyr (amino acids 36–357, NCBI Reference Sequence WP_009958178.1) was PCR-amplified from pETHis–protyr using T7 promoter primer and a reverse primer 5′ [TTT TTT GGT ACC TTA AAC TGC TTC ACC GCT ACC TGG] 3′. The amplified product was then cloned into the *SspI–KpnI* site of a pETHis–LIC cloning vector (Addgene, Watertown, MA, USA) to obtain pETHis-ctyr. For the expression of pro-*vs*Tyr and the core domain of *vs*Tyr, the above plasmids (pETHis–protyr and pETHis-ctyr) were transformed into *E. coli* BL21(DE3) pLysS (Promega, Nacka, Sweden) for expression in shake flasks containing 700 mL of terrific broth (TB) media supplemented with 50 μg/mL ampicillin and 50 μg/mL chloramphenicol. The growth was carried out at 37 °C and 200 rpm until the OD_600 nm_ reached 3.0. The cells were induced with 1 mM isopropyl β-D-thiogalactoside (IPTG) and expression was carried out overnight at 30 °C, 200 rpm. Cells were harvested by centrifugation and subsequently frozen at −80 °C before further processing.

The core domain of *vs*Tyr was purified by first resuspending the expressed cell pellets in 5 mL of lysis buffer (20 mM NaH_2_PO_4_ pH 7.4, 500 mM NaCl, 20 mM imidazole, 1 cOmplete protease inhibitor tablet, 10 mg lysozyme, 1 mg DNaseI and 2 mM MgCl_2_) per g cells (wet weight). Samples were lysed by sonication and centrifuged at 40,000× *g* for 1 h at 4 °C. The soluble cell lysate was collected and subjected to immobilized-metal ion affinity chromatography using an Äkta Pure 150 M system. A 5 mL HisTrap FF column (Cytiva, Uppsala, Sweden) charged with Ni^2+^ was equilibrated with 5 column volumes of binding buffer (20 mM NaH_2_PO_4_ pH 7.4, 500 mM NaCl, 20 mM imidazole) before loading the lysate on the column. It was subsequently washed with 5 column volumes of binding buffer and the protein was eluted by a gradient of 20–350 mM imidazole in the binding buffer. Fractions with tyrosinase activity were pooled and desalted with 10 mM Tris-HCl buffer (pH 8.0) using a PD10 desalting column packed with Sephadex G-25 resin (Cytiva, Uppsala, Sweden). The N-terminal His_6_-tag of the purified protein (2 mg/mL) (See Appendix A) was cleaved off by overnight incubation with tobacco etch virus (TEV) protease (1 mg/mL) at 4 °C. The core domain of *vs*Tyr was separated from TEV and un-cleaved His-tagged *vs*Tyr using an IMAC step. The flow-through, containing the tag-free *vs*Tyr core domain, was subsequently desalted using 10 mM Tris-HCl buffer (pH 8.0). The pro-*vs*Tyr was purified using the above-mentioned procedure. For some experiments, it was trypsinized by incubation with 10 µg trypsin at room temperature at low shaking for 18 h.

The *vs*Tyr core domain and trypsinized pro-*vs*Tyr were reconstituted with copper by mixing the protein with a threefold molar excess of CuSO_4_. The sample was incubated on ice for 1 h and then desalted using a PD10 column in 10 mM Tris-HCl buffer (pH 8.0). The reconstituted *vs*Tyr core domain protein was concentrated to 30 mg/mL for crystallization.

### 2.2. vsTyr Enzyme Activity Assay

The tyrosinase activity of *vs*Tyr was measured using L-DOPA (l-3,4-dihydroxyphenylalanine) or L-tyrosine as a substrate in a reaction mixture containing 25 mM potassium phosphate buffer, pH 6.8, 1 mM substrate, 0.01 mM CuSO_4_ and varied concentrations of the enzyme. The production of DOPAchrome was monitored at 475 nm using UV/Vis spectrophotometry over a 1 min time course at 30 °C, and quantified with a molar extinction coefficient of 3600 M^−1^∙cm^−1^ [30].

### 2.3. Crystallization of vsTyr Core Domain

Screening for the initial crystallization conditions was performed at 18 °C by sitting drop vapor diffusion in 96-well plates using sparse-matrix screens (Molecular Dimensions, Rotherham, UK). Crystals appeared within 3 days under several conditions, which were independently optimized. The crystals used for data collection were obtained by equilibrating 1 µL droplets against a reservoir containing 0.1 M (NH_4_)_2_SO_4_, 0.1 M MES pH 6.5, and 30% *w*/*v* PEG5000-MME, at 20 °C. The drops consisted of equal volumes of protein (20 mg/mL) and reservoir solution. Crystals were cryo-protected by a quick plunge in reservoir solution supplemented with 20% (*v*/*v*) glycerol (cryo-solution), and flash-frozen in liquid nitrogen. After the initially obtained structure revealed low occupancy of the metal-binding sites, crystals used in subsequent experiments were, prior to freezing, soaked for varying time periods in a cryo-solution containing 1 mM CuSO_4_ followed by a quick wash in CuSO_4_-free cryo-solution. Soaking times of ca. 15 min resulted in the highest observed metal site occupancy.

To obtain ligand complexes, the *vs*Tyr core domain was incubated with either L-tyrosine, L-DOPA, Tyr-containing peptide substrates or the inhibitor kojic acid for several hours prior to crystallization using the above-mentioned conditions or crystallization screens were set up to search for new conditions. Furthermore, ligand-free crystals of the *vs*Tyr core domain were soaked with these ligands for different time periods, up to overnight.

### 2.4. Data Collection, Structure Determination and Refinement

All crystallographic data were collected in unattended mode, monitored by SynchWeb [31] at beamline I03 of the Diamond Light Source (DLS, Didcot, UK). Datasets were processed by autoPROC [32], data reduction and scaling were carried out using POINTLESS [33] and AIMLESS [34], respectively. The MRage molecular replacement pipeline [35,36] was used to identify suitable search models (PDB accession codes 6ELS, 5CE9 and 4Z12) and to perform the initial molecular replacement search. Density modification was performed using Phenix AutoBuild [37], followed by interactive modelling in Coot [38,39] to adjust for poorly fitted parts of the initial molecular replacement solution. The improved model was used for iterative density modification, chain rebuilding and extension, as well as for refinement in AutoBuild. Coot and Refmac5 [40] were applied for the final cycles of model-building and refinement. The occupancies of the copper ion binding sites were estimated and refined based on their anomalous contributions to the diffraction data. The data collection and refinement statistics are given in Table 1 and Appendix A. All figures were prepared using the PyMol Molecular Graphics System (Version 2.5.2, Schrödinger, LLC).

### 2.5. Prediction of the Full-Length vsTyr Structure Using AlphaFold

The three-dimensional structure of pro-*vs*Tyr was predicted using AlphaFold [41,42]. AlphaFold (version 2.0), obtained from https://github.com/deepmind/alphafold (accessed on 17 August 2023) was run locally using a script from https://github.com/kalininalab/alphafold_non_docker (accessed on 17 August 2023), which was adapted for the local setup. The full-length sequence of pro-*vs*Tyr (amino acids 1–518, NCBI Reference Sequence WP_009958178.1) was obtained in FASTA format and used for prediction. The multiple sequence alignment (MSA) for the sequence was performed on the full genetic databases for the structure predictions. Other run parameters were kept as default parameters. The generated structures (default = 5) were subjected to Amber relaxation as implemented in AlphaFold and were further ranked based on the average predicted local difference distance test (pLDDT), a measure of local structural accuracy. The best-ranked model had a score of 88.96 and was used for further analysis in the study. Another metric for the reliability of predictions is the predicted aligned error (PAE), which is a measure for the confidence in the relative positions of two residues in the modelled structure. The PAE plot for the best model was generated using PAE Viewer [43].

**Table 1 biomolecules-13-01360-t001:** Data collection and refinement statistics ^a^.

	Unsoaked Crystals	CuSO_4_-Soaked Crystals
**Data Collection**		
Space group	C2	C2
a, b, c (Å), β (°)	84.8, 63.4, 116.0, 96.8	86.3, 63.4, 117.3, 97.2
Molecules in a. u.	2	2
Wavelength (Å)	0.97626	0.97625
Resolution (Å) ^b^	115.20–1.43 (1.56–1.43)	116.37–1.64 (1.67–1.64)
Total reflections	426,619 (19,225)	508,362 (15,210)
Unique reflections	62,664 (3134)	75,699 (3073)
Multiplicity	6.8 (6.1)	6.7 (4.9)
Completeness (%)	93.4 (66.8)	98.2 (79.0)
˂I/σ(*I*)˃	10.5 (1.6)	9.8 (0.6)
Wilson B factor (Å^2^)	23.5	25.8
R*_merge_*	0.104 (1.084)	0.114 (2.182)
R*_meas_*	0.112 (1.182)	0.124 (2.441)
R*_pim_*	0.043 (0.463)	0.047 (1.063)
* CC_1/2_ *	0.99 (0.60)	0.998 (0.362)
**Refinement**		
No. of reflection in work set	62,663	77,117
No. of reflection in free set	7016	3975
R*_work_*	0.1696	0.1907
R*_free _*^c^	0.2017	0.2245
No. of non-hydrogen atoms		
protein	5226	5209
solvent	517	705
Cu^2+^	4	13
RMS deviations		
bonds (Å)	0.008	0.007
angles (°)	1.488	1.503
Ramachandran plot regions (% residues)		
favored	98.6	98.7
allowed	0.8	0.6
outliers	0.6	0.6
Average B factor (Å^2^)		
protein	16.3	27.1
solvent	27.3	36.0
Cu^2+^	24.3	49.6
PDB-ID	8BBQ	8BBR

^a^ The statistics were obtained from AIMLESS [34], Refmac5 [40], Baverage [44], and Rampage [45]. ^b^ Values in parentheses are for the highest resolution shell. ^c^ R_free_ is calculated for 5% reflections not used in the refinement.

## 3. Results and Discussion

### 3.1. Production and Activity of vsTyr Variants

The bacterium *Verrucomicrobium spinosum* produces its tyrosinase as a pre-pro-enzyme whose catalytic activity is attenuated by a C-terminal extension. We initially produced and analyzed the activity of both the pro-enzyme (here denoted as “full-length”) and the fully processed enzyme without the C-terminal extension (known as the “core domain”, representing amino acids 36–357). The recombinant expression of full-length pro-*vs*Tyr was low compared to the core domain *vs*Tyr. The protein was monomeric, as judged from the size-exclusion chromatography analysis (not shown).

The activities of both produced forms of the enzyme were compared to determine if (and by how much) the C-terminal domain affected its activity. The specific activity of the core domain was considerably higher with L-DOPA than that of full-length *vs*Tyr, 213 U/mg vs. 0.67 U/mg (1 U is defined as 1 µmol DOPAchrome produced per minute). A comparison of the activity of the purified and trypsinized pro-*vs*Tyr with the purified core domain of *vs*Tyr showed that the latter had much higher activity with both L-DOPA and L-tyrosine than the trypsinized pro-*vs*Tyr (see Appendix A and Appendix A for details). Moreover, the purified, trypsinized pro-*vs*Tyr was slightly more stable, as it retained 100% activity upon storage at 4 °C for 3 weeks, while the purified core domain of *vs*Tyr retained approximately 87% of its activity under the same conditions and time periods. However, since we were primarily interested in exploring the most catalytically active form of *vs*Tyr, the structural work was focused on the core domain. It was produced directly rather than indirectly by the trypsinization of pro-*vs*Tyr.

### 3.2. Structure of the vsTyr core Domain

Crystals of the *vs*Tyr core domain belong to the space group C2 containing two monomers per asymmetric unit. Despite reconstitution with CuSO_4_ after its purification, the initial crystal structure, determined by molecular replacement at 1.43 Å resolution, revealed that the copper sites were only partially occupied (Appendix A). A second dataset was therefore collected from a *vs*Tyr crystal soaked with CuSO_4_ for 15 min (1.64 Å), resulting in considerably improved metal site occupancy. Data collection and refinement statistics for both structures are given in Table 1 and Appendix A. The final models contain all amino acids of the crystallized construct, except for the C-terminal two (unsoaked) and three amino acids (soaked) in chains B, respectively. The models also include additional 1–2 amino acids derived from the purification tag at the N-terminus, for which electron density was observed.

Despite the low overall sequence identity of <30% with other type-III copper enzymes (Table 2; note that higher identities are observed for smaller parts of the sequences whose extents are indicated by the %Seq cov values), *vs*Tyr adopts the typical core fold of this protein family characterized by a central 4-helix bundle [46]. Additionally, the peripheral regions of the *vs*Tyr core domain are almost exclusively comprised of α-helices and 3_10_-helical elements (Figure 2 and Figure 3). Two short β-strands are found at the beginning and end of the polypeptide chain, respectively, together forming a small sheet. The active site with the di-copper center is solvent-accessible, being located at the midpoint of the core domain.

In general, the fold of the domain is very compact, which likely contributes to the high structural stability [28]. B-factor analyses of both deposited models revealed that the N- and C-termini are the most flexible parts of the subunit, and that there is little main chain atom B-factor variation across the sequence except for a modest increase in three solvent-exposed regions (86–87, 176–193, 300–305). For at least two of these regions, the inherent flexibility indicated by the higher B-factors may be of functional importance, e.g., in enzyme-loading with Cu^2+^ ions, as they encompass elements of the metal-binding sites.

Superimposition of *vs*Tyr with other PPO structures identifies *Vitis vinifera* tyrosinase (*vv*Tyr, PDB ID: 2P3X [25]) and *Ipomoea batatas* catechol oxidase (*ib*CO, 1BT1 [21]) as the closest structural homologues, closely followed by *Coreopsis grandiflora* aurone synthase (*cg*AUS, 4Z11 [48]) and other plant, fungal, and bacterial tyrosinases (PDB IDs: 5CE9 [20], 6ELS [49], 5ZRD [26]). It is primarily the central 4-helix bundle of the cTyr core domain that superimposes well with homologous enzyme structures (Figure 4 and Appendix A), including the positions of the copper-ligating histidine residues and other catalytic residues harbored by it. Other structural features are unique to *vs*Tyr, such as the arm-like structure formed by α-helices α4-α6 and connecting loops that is “hugging” the surface of the core domain.

The root mean square deviations (RMSDs), number of aligned Cα atoms, and other significant characteristics of the homologous enzymes are given in Table 2. They show that *vs*Tyr cannot clearly be grouped within a particular phylogenetic group of these enzymes, as it shares the presence of the C-terminal extension with both plant and fungal tyrosinases, as well as with *Burkholderia thailandensis* tyrosinase (*bt*Tyr), the only other bacterial tyrosinase of a known structure containing this inhibitory domain. See Appendix A for a phylogenetic tree for the proteins mentioned in Table 2 and their evolutionary relation to *vs*Tyr (8BBQ).

**Table 2 biomolecules-13-01360-t002:** Comparison of sequence and structural characteristics of *vs*Tyr and the most closely related type-III copper enzyme family members of a known structure. The highest sequence and structural similarities, and the Cys–His thioether linkages with the same topology of *vs*Tyr are highlighted (bold). One pair of activity controller residues and one of the gate keeper residues matches those in *vs*Tyr (bold boxes).

Enzyme Source Type	PDB-Id [Ref]	Enzyme Name (Organism), Abbreviation	Sequence Similarity ^a^ (% Seq-Id/ % Seq Cov/ E)	Structural Similarity ^b^ (RMSD, Å/ #Aligned Cα Atoms)	Activity Regulated via C-Term Ext or Caddie Protein	Cys–His Thioether Linkage Yes/No, (Topology Relative *vsTyr*)	Activity Controller Residue 1 (HisB1 + 1)	Activity Controller Residue 2 (HisB2 + 1)	Gatekeeper Residue	Additional Gatekeeper Residue ^c^
Bacterial	8BBQ	**Tyrosinase** (*Verrucomicrobium spinosum*), *vs*Tyr	Reference sequence	Reference structure	C-term ext	Y	**Asn259**	**Asn263**	**Leu272**	**Phe453**
	5ZRD [26]	**Tyrosinase** (*Burkholderia thailandensis*), *bt*Tyr	22.4/41/4 × 10^−8^	1.96/218	C-term ext	**Y (same)**	**Asn277**	Leu281	Asn306	Leu466
	3NM8 [8]	**Tyrosinase** (*Bacillus megaterium*), *bm*Tyr	28.6/42/5 × 10^−9^	2.11/213	No caddie	N	**Asn259**	His209	Val218	-
	6J2U [Not publ.]	**Tyrosinase** (*Streptomyces avermitilis*), *sa*Tyr	47.2/10/5 × 10^−6^	2.27/208	Caddie	N	**Asn191**	Val195	Gly204	Tyr92
	7CIY [50]	**Tyrosinase ** (*Streptomyces castaneoglobisporus*), *sc*Tyr	46.3/7/ 3 × 10^−5^	2.29/212	Caddie	N	Gly191	Val195	Gly204	Tyr*98 (dopaquinone)
Plant	4Z11 [48]	Aurone synthase (*Coreopsis grandiflora*), *cg*AUS	25.8/62/**4 × 10^−19^**	1.77/224	C-term ext	Y	Thr256	Arg260	Phe273	Ile456
	6HQI [51]	Polyphenol oxidase 1 (*Solanum lycopersicum*), *sl*Tyr	21.0/91/4 × 10^−11^	2.09/240	C-term ext	? (not resolved)	Ser242	Ile246	Phe270	Leu447
	1BT1 [21]	Catechol oxidase (*Ipomoea batatas*), *ib*CO	32.6/36/9 × 10^−10^	1.67/223	C-term ext [52]	Y	Ile241	Arg245	Phe261	-
	2P3X [25]	**Tyrosinase** (*Vitis vinifera*), *vv*Tyr	22.6/57/8 × 10^−13^	**1.66**/217	C-term ext	Y	**Asn240**	Lys244	Phe259	-
	5CE9 [20]	**Tyrosinase** (*Juglans regia*), *jr*Tyr	25.8/52/2 × 10^−18^	1.69/223	C-term ext [53]	Y	**Asn240**	Leu244	Phe260	-
	6ELS [49]	**Tyrosinase** (*Malus domestica*), *md*Tyr	24.3/65/1 × 10^−16^	2.01/240	C-term ext	Y	Ala239	Leu243	Phe259	Val449
Fungal	2Y9W [54]	**Tyrosinase** (*Agaricus bisporus*), *ab*Tyr	23.4/61/2 × 10^−4^	1.93/212	C-term ext	**Y (same)**	**Asn260**	Phe264	Val283	-
	3W6Q [55]	**Pro-tyrosinase** (*Aspergillus oryzae*), *ao*Tyr	41.8/7/0.001	2.63/270	C-term ext	**Y (same)**	**Asn329**	**Asn333**	Val359	**Phe513**
	6Z1S [56]	Tyrosinase-like protein (*Thermothelomyces thermophila ATCC 42464*), *tt*TyrP	25.9/48/1 × 10^−9^	2.17/224	No C-term ext	N	**Asn292**	Tyr296	Phe307	-
	4OUA [57]	**Tyrosinase** (*Agaricus bisporus var. bisporus H97*), *abb*Tyr	21.4/38/3 × 10^−6^	2.18/227	C-term ext	**Y (same)**	Asp252	Gly256	**Leu271**	**Phe454**
Human	5M8L [58]	Tyrosinase-related protein 1 (*Homo sapiens*), *hs*TyrRP1	22.9/45/2 × 10^−5^	2.36/208	-	N	**Asn378**	Leu382	Thr391	-

^a^ Sequence alignments performed with BLAST. ^b^ Structural alignments performed with LSQkab implemented in CCP4 [44]. ^c^ Residues in structures in which they are present/visible.

### 3.3. Di-Copper Center

The active site contains a metal center consisting of two copper ions (CuA and CuB) bridged by a water (or hydroxyl) molecule. During the reaction catalyzed by *vs*Tyr, this water would initially be replaced by a dioxygen molecule. The electron density map features of the metal site do not allow for unambiguous identification of the bridging moiety; nevertheless, the distance between the A and B sites is a functionally relevant feature in type-III copper enzymes correlated with the oxidation state of the metal center. For *vs*Tyr, the distances of 3.7 Å and 4.0 Å observed in the structures determined from unsoaked and CuSO_4_-presoaked crystals, respectively, fall within the 3.2–4.0 Å range that is characteristic for the met form. Shorter distances would be characteristic for the oxy form, whereas longer distances are associated with the deoxy form [59]. This is compatible with PPOs which were mainly characterized in their resting met form, in which a water molecule or hydroxide ion bridges the two copper ions in the cupric state [10].

Each copper ion is coordinated by three histidine residues, CuA by His80, His86 and His95, and CuB by His258, His262 and His284 (Figure 5A and Appendix A), of which all but one originate from the four helices (α2, α3, α11, α12) forming a bundle around the metal center. The exception is His86, located on the loop connecting α2 and α3. The main chain atoms in this loop exhibit higher B-factors than neighboring residues or other not fully solvent-exposed loop regions, indicating flexibility. It has previously been suggested, for *sc*Tyr [60], that the flexibility of this loop is of functional importance as it may facilitate copper ion incorporation into the active site.

Another feature setting His86 apart is its covalent linkage to Cys84 via a thioether bond (Figure 5A). Such a His–Cys link is also found in other copper-containing enzymes, e.g., *ib*CO, *ao*Tyr, and *molluscan* hemocyanin [11,14,21,54,61] (Figure 5B), but is, with the exception of *bt*Tyr, absent in other bacterial tyrosinases (Figure 5C). The covalent connection with the cysteine is thought to stabilize the position of the copper-ligating histidine, and consequently also the position of CuA, and likely plays a role in fine-tuning the redox potential of the enzyme for fast electron transfer [21]. Interestingly, the cysteine approaches His86 (generally termed HisA2) from another side than the corresponding cysteines in plant-derived tyrosinases and homologous PPOs, which have a longer loop inserted between the two covalently linked residues. Only in *bt*Tyr and fungal tyrosinases is the positioning of the cysteine identical to that in *vs*Tyr (Figure 5B).

Although the presence of the Cys–His link is supported by the electron density map (Appendix A), the relative weak density observed for the His86 side chain (especially around Cβ) indicates that it may not be formed in all *vs*Tyr molecules comprising the crystals, allowing His86 and Cys84 to occupy alternative positions placing them slightly more apart from each other. Whether this is of functional relevance, or is an artifact caused by the experimental conditions is currently unclear.

An additional histidine well conserved in bacterial, fungal and plant PPOs [57,62] is found near CuB. In *vs*Tyr, this histidine corresponds to His283, but in neither of the two structures reported here is it directly coordinating this ion, nor is it hydrogen-bonded to a water molecule or residues that may be involved in catalysis (Figure 5 and Appendix A). Instead, it forms hydrogen bonds to the side chain of Trp122 and the carbonyl oxygen of the copper-ligand His258, and hence may play a role in maintaining active site geometry and CuB site stability. This does not, however, exclude a role as an alternative copper-ligand during the catalytic cycle, which may depend on a certain plasticity of the copper-binding sites [10,46]. Mutagenesis studies performed for PPO have shown that replacement of this histidine results in reduced copper content and partial or complete inactivity of the mutant enzymes [10].

### 3.4. Copper Site Occupancy and Additional Copper Binding Sites

Despite reconstitution of the purified *vs*Tyr core domain with Cu^2+^ ions prior to crystallization, the initially determined crystal structure indicated low occupancy of the metal-binding sites. Using the anomalous contribution of the Cu ions, occupancies of the CuA and CuB sites for the two copies of *vs*Tyr per asymmetric unit could be estimated to be ~40% and ~80%, respectively (Appendix A). The lower occupancy observed for the A site may be caused by its proximity to His86, which belongs to a flexible loop. The *vs*Tyr structure, determined from a crystal presoaked with CuSO_4_ for 15 min, revealed significantly increased metal site occupancies of 70–85% for site A and >90% for site B, respectively (Appendix A). Interestingly, the increased metal site occupancy is the only noticeable difference between the two *vs*Tyr core domain structures reported here. Their superposition results in a very low RMSD of 0.4 Å and identical crystal packing, hinting at the remarkable structural stability of the *vs*Tyr core domain.

Previous studies have shown that the occupancies of the two catalytic copper-binding sites vary for some tyrosinases and that their exact positions show some flexibility. In *sc*Tyr, a tyrosinase whose active site is supplied with copper ions by a caddie protein, CuB, showed higher positional stability and occupancy than CuA [60], which was attributed to the proximity of CuA to the flexible ligating histidine that may have a functional implication [14]. The authors suggested that the CuB site is supplied with Cu^2+^ first, followed by CuA. In *bm*Tyr, a bacterial tyrosinase not requiring a caddie protein for activation, the site occupancy varied depending on the crystallization and soaking conditions, with one structure showing an empty CuB site [8]. In *Agaricus bisporus* tyrosinase (*ab*Tyr), the CuA site was found to be well defined, whereas CuB showed a slight positional variation [57]. Thus, our results are in line with the previously observed behavior of different types of tyrosinases.

Crystal soaking led to the appearance of anomalous density peaks on the protein surface, but outside the active site, that could be interpreted as additional Cu^2+^-binding sites (Appendix A). One ion is bound by His39 and two are bound by Glu108 in both *vs*Tyr monomers present in the asymmetric unit. A seventh ion is bound by Asp344 in chain B only. In the *vs*Tyr crystals, this last site is located directly adjacent to the same site found in a neighboring symmetry mate. These additional binding sites are most likely not of any physiological relevance, since their affinities for Cu^2+^ are expected to be low, and none of them has to date been observed in homologous enzymes.

### 3.5. The Waterkeeper Residue

The catalytic cycle of tyrosinases is initiated by binding of the monophenolic substrates and their deprotonation to phenolates. The catalytic base in this reaction was originally proposed to be a water molecule bound at a conserved position near the binuclear copper center, between an asparagine residue and a “waterkeeper” glutamate supposed to increase its basicity by proton abstraction [63]. Both residues are found in almost all biochemically characterized PPOs [55,64], including *vs*Tyr (Glu254 and Asn259, Figure 4D). Mutagenesis of this Glu in homologous enzymes to an uncharged or a positively charged amino acid resulted in a loss of activity [65], confirming its essential role for both the mono- and di-phenolase activity of diverse PPOs that is most likely linked to the stable positioning or activation of water molecules. Nevertheless, Kampatsikas et al. [51] more recently assigned the function of the base deprotonating incoming monophenolic substrates to one or two copper-ligating histidine residues (HisB1, HisB2), which are supported in this role by so-called activity controller residues.

### 3.6. Activity Controller Residues

During site-directed mutagenesis studies aimed at installing tyrosinase activity in an aurone synthase (*cg*AUS), Kampatsikas et al. [10,51] identified two residues that seemed to be largely responsible for the ability or inability of diverse PPOs to convert monophenols into di-phenols. Highest monophenolase activity could be achieved with a *cg*AUS mutant containing aspartate or asparagine at the positions HisB1+1 and HisB2+1, i.e., following the first and second CuB-ligating histidine residues. These positions are generally not conserved in type-III copper enzymes, but with just two exceptions, all structurally characterized tyrosinases including *vs*Tyr (Asn259) do indeed carry an asparagine or aspartate at position HisB1+1 (see Table 2), which is hydrogen-bonded (~2.8 Å) to its neighboring histidine and can thus stabilize the position and conformation of the CuB-ligand and increase its basicity to facilitate its potential role in the deprotonation of monophenolic substrates.

At position HisB2+1, denoted as a “second activity controller” [10,51], significant variation in the amino acid type is observed even among tyrosinases. For *cg*AUS, the mutation of the corresponding Arg257 to aspartate significantly contributed to the generation of monophenolase activity in this enzyme [57]. Kampatsikas et al. hypothesized that this aspartate would play a similar activating role for the second CuB-ligating histidine as the first activity controller does for the HisB1 but may also have a stronger substrate guiding effect. Besides *ao*Tyr, *vs*Tyr is the only other structurally characterized PPO which contains an asparagine (Ans263) at position HisB2+1, whereas none natively contains an aspartate. However, though its side chain is oriented towards His262, Asn263 is not hydrogen-bonded to it, nor would a simple conformer change of either residue allow for the formation of such a bond without necessitating larger structural rearrangements of other nearby residues. In *ao*Tyr, the corresponding Asn333 is also not hydrogen-bonded to HisB2 (His333) and is also oriented away from this position due to the insertion of Phe511 of the C-terminal extension in the space that Asn333 would otherwise occupy. It is thus unlikely that the second activity controller is playing a role in HisB2 activation, at least in these two enzymes. Its “activity controlling” role may instead be linked to substrate guidance effects, or (regulation of) the interactions with the C-terminal blocker domain, or both.

### 3.7. The Gatekeeper Residue

Type-III copper enzymes usually contain a so-called gatekeeper residue blocking access to the di-copper center from above, thus controlling enzymatic activity. In enzymes derived from plants this is a bulky aromatic residue, typically phenylalanine [61,66]. It has been suggested that the gatekeeper inhibits the undesirable oxidation of phenolic compounds and was thought to be the reason for the lack of monophenolase activity in catechol oxidases (COs) [21,67]. However, recent mutagenesis studies on a CO revealed that the lack of monophenolase activity can largely be attributed to the absence of amino acid side chains that can increase the basicity of crucial Cu^2+^-ligating histidine residues that may double-function in substrate deprotonation [20,51,65,68,69]. The gatekeeper residue may instead either inhibit or help substrates to enter the active site [10], with the thioether bridge playing a role in restraining its flexibility and thereby stabilizing its position [51].

In the crystal structure of the *vs*Tyr core domain, the conserved location of the side chain of the gatekeeper is unoccupied, and the active site is thus freely accessible. This is because the side chain of Leu272 occurring at the corresponding position in sequence is smaller and differently placed (Figure 6). *vs*Tyr is not unique in this respect, almost all type-III copper enzymes of non-plant origin contain a smaller and less bulky residue as a gatekeeper, e.g., asparagine, valine, threonine or even glycine. Interestingly, in the fungal tyrosinases *ao*Tyr (PDB-Id 3W6Q) and *ab*Tyr (4OUA), and in the bacterial tyrosinases from two Streptomyces species (6J2U, 7CIY), phenylalanine, or tyrosine side chains occupy the same position as the gatekeeper phenylalanine residues of plant PPOs, though approaching it from the opposite direction (Figure 6). The phenylalanine residues, Phe513 in *ao*Tyr and Phe454 in *ab*Tyr, belong to the C-terminal extension, whereas the tyrosine residues of *sa*Tyr (Tyr92) and *ac*Tyr (Tyr98) belong to their respective caddie proteins that were co-crystallized with the enzymes. Sequence alignment with other PPOs previously identified Phe453 from the C-terminal extension as *vs*Tyr´s potential gatekeeper [27]. Its mutation to alanine led to an increase in catalytic activity compared to wild-type *vs*Tyr [27], indicating that Phe453 is indeed the gatekeeper, and that its primary role is to block, rather than facilitate, substrate entry to the active site. To further investigate this, we superimposed the structure of full-length *vs*Tyr predicted by AlphaFold onto the core domain crystal structure (RMSD: 0.495 Å, Appendix A). As observed for the corresponding phenylalanine residues in *ao*Tyr and *ab*Tyr, the space predicted to be occupied by the side chain of Phe453 overlaps with that occupied by the conserved gatekeepers in plant PPOs (Figure 6A).

It is thus possible that in fungal and bacterial PPOs, the active site-blocking function of the gatekeeper is “outsourced” to a second position residing in the C-terminal extension or the caddie protein. The *vs*Tyr Phe453Ala variant also exhibited significant underloading of the copper sites and, in contrast to the full-length native *vs*Tyr, an ability to readily form the active oxy complex, suggesting additional roles of the “outsourced gatekeeper” in maintaining the integrity of the binuclear metal center and the regulation of oxygen binding to the enzyme. Whether a substrate guidance function remains associated with the originally identified gatekeeper position in these enzymes requires further investigation.

AlphaFold predicted a β-jelly roll fold for the *vs*Tyr C-terminal extension, which is thus structurally very similar to the corresponding extensions of *cg*AUS, *ml*Tyr, and *sl*Tyr (Appendix A). Interestingly, a structure similarity search using the DALI server revealed that the next closest homologous domains are found in a variety of carbohydrate-binding or -processing proteins. Whether tyrosinase C-terminal extensions also function in carbohydrate recognition has, to our knowledge, not yet been investigated.

In plant PPOs crystallized with their C-terminal extension present (*cg*AUS, *sl*Tyr, *md*Tyr), the alternative gatekeeper position is occupied by an aliphatic hydrophobic residue (Table 2, Figure 6B), whose functional importance has, to our knowledge, not yet been tested via site-directed mutagenesis studies. It is furthermore noteworthy that *bt*Tyr is currently the only structurally characterized PPO that does not carry an aromatic side chain at either one of the two gatekeeper positions.

### 3.8. Crystallisation Trials with vsTyr in Complex with Ligands

Our attempts to obtain a crystal structure of the core domain of *vs*Tyr with substrates (L-tyrosine, L-DOPA and tyrosine-containing peptides) or the inhibitor kojic acid bound to the active site remained unsuccessful, with both co-crystallization and crystal soaking setups, irrespective of whether the experiments were performed in the absence or presence of CuSO_4_. No ligand-associated electron density could be detected in any of the tested crystals, indicating that the active site of the *vs*Tyr core domain has low affinity for these ligands under the explored conditions. Comparable experiments have also been unsuccessful for other tyrosinases, e.g., *bm*Tyr, suggesting that this is a general feature of tyrosinases [8].

## 4. Conclusions

The crystal structure of the active core domain of *vs*Tyr shows that the active site and copper coordination residues were identical to those in other type-III copper enzymes, despite a low overall sequence identity with any of the homologous enzymes. Each of the two copper ions is coordinated by three histidine residues, of which one, His 86, forms a covalent bond with cysteine 84. This thioether link is a common feature for eukaryotic tyrosinases but is rarely found in enzymes of bacterial origin. Furthermore, we identified Glu254 as the waterkeeper residue, and Asn259 and Asn263 as activity controllers, making *vs*Tyr, together with a fungal tyrosinase, the only structurally characterized tyrosinases to date that carry asparagine at both activity-controlling positions. By using AlphaFold for the prediction of the full-length *vs*Tyr structure we could assign the function of (primary) gatekeeper to Phe453, which resides in the C-terminal extension domain not included in the crystal structure. This is yet another feature that makes *vs*Tyr rather unique among bacterial tyrosinases, which generally do not contain a C-terminal domain and outsource the gatekeeper position to a separate accessory (caddie) protein. The primary function of this gatekeeper appears to be pro-enzyme inhibition by the prevention of substrate access to the active site. Enzyme activation is achieved by the removal of the C-terminal extension from the pro-enzyme by proteolytic cleavage, with the activated enzyme consisting of the remaining core domain.

Our finding that the tyrosinase of *Veruccomicrobium spinosum* shares more catalytic and structural features with homologous enzymes outside its own phylogenetic group than with other bacterial tyrosinases raises the question of its evolutionary origin. The fact that it *combines* the characteristics of plant, fungal and bacterial tyrosinases would suggest that it is not the result of lateral gene transfer. Interestingly, tyrosinase is not the only example of a *Veruccomicrobium* protein with traits of eukaryotic origin. These bacteria do, for instance, contain tubulin, which is found in all eukaryotes as a component of the cytoskeleton, but rarely in bacteria, and in contrast to other bacterial tubulins, it has been argued against being the result of lateral gene transfer from a eukaryote due to its distinct biochemical characteristics [70]. Thus, the tyrosinase only adds to the mysteries surrounding the evolution of the PVC superphylum, composed of the bacterial phyla *Planctomycetes*, *Verrucomicrobia, Chlamydiae* (PVC) and other species of related ancestry but very diverse pheno- and genotypes [71].

## Figures and Tables

**Figure 1 biomolecules-13-01360-f001:**
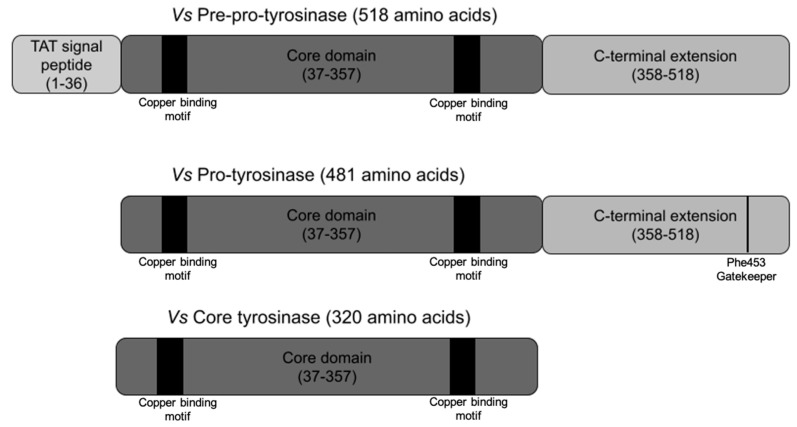
The domain structure of the tyrosinase from the bacterium *Verrucomicrobium spinosum* (*vs*Tyr) with the delimitations of each domain indicated by the amino acid numbering. See Appendix A for the exact sequence used in this study.

**Figure 2 biomolecules-13-01360-f002:**
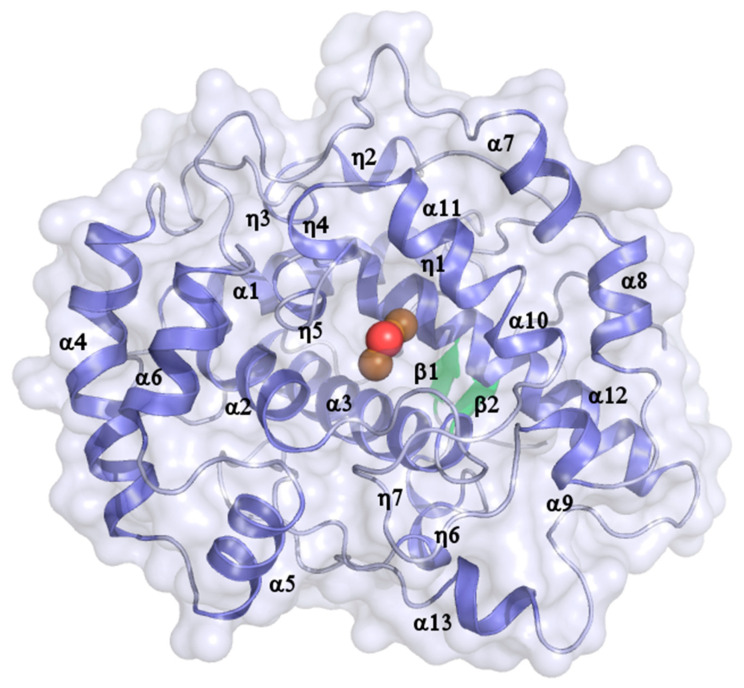
Crystal structure of the core domain of *vs*Tyr. Secondary structure elements are numbered consecutively for each type, with β labeling β-strands, and α and η denoting α- and 3_10_-helices, respectively. The two copper ions of the di-metal center (orange) and the bridging water molecule (red) are shown as spheres. The molecular dimensions are outlined by a semi-transparent surface, which highlights the solvent accessibility of the active site in the absence of the C-terminal extension.

**Figure 3 biomolecules-13-01360-f003:**
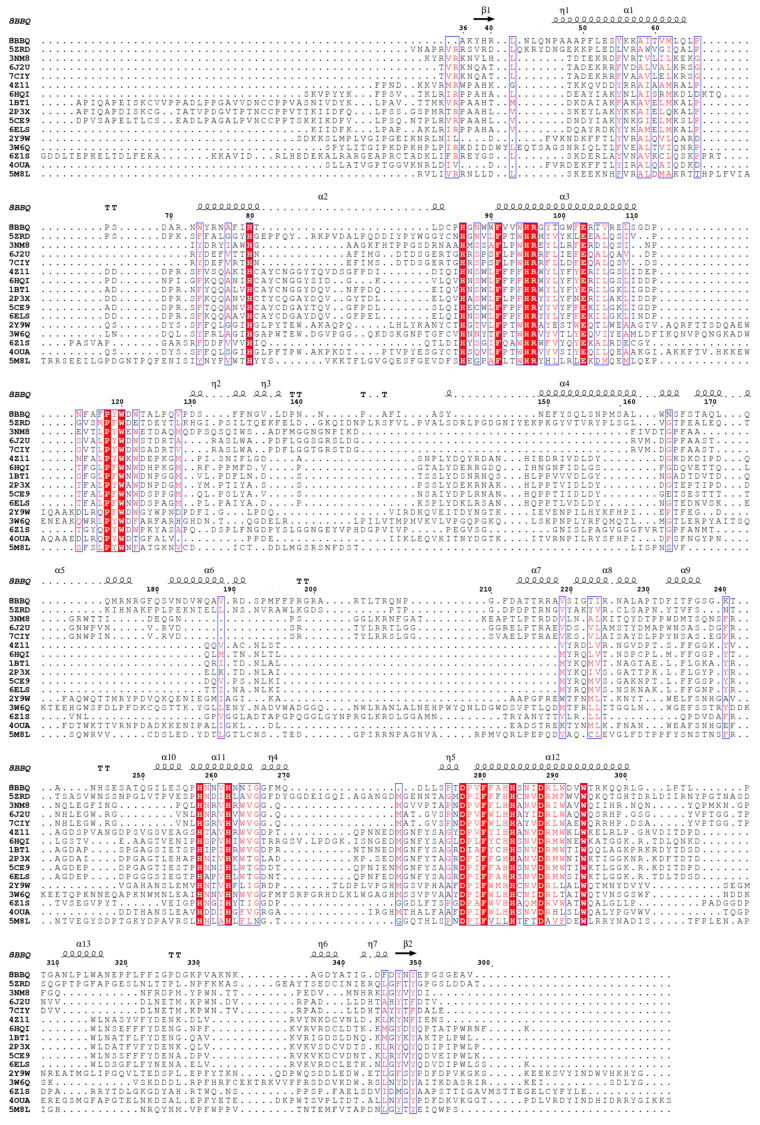
Multiple amino acid sequence alignment of the core domains of *vs*Tyr (8BBQ) and homologous proteins using the amino acid sequences of the respective PDB entries (the sequences are labelled with the corresponding accession codes, see Table 2 for full names). The secondary structure elements of *vs*Tyr are indicated above the alignment, with arrows representing β-strands (β), squiggles α- (α) and 3_10_-helices (η), and TT strict β-turns. White characters in red boxes indicate amino acid conservation in all aligned sequences, red characters indicate identity or similarity within a group, and blue frames indicate similarity across sequence groups. The alignment was produced using ESPript 3.0 [47].

**Figure 4 biomolecules-13-01360-f004:**
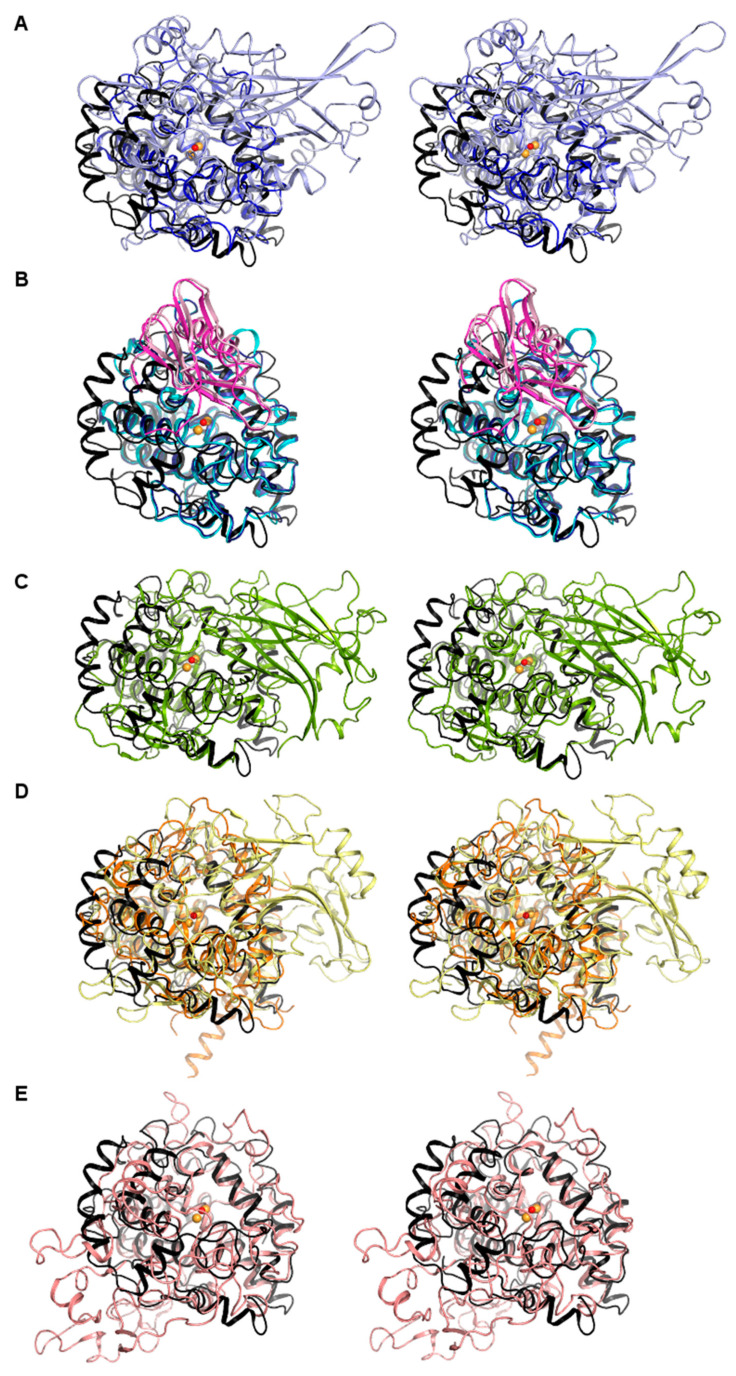
Stereo views of structural superimpositions of *vs*Tyr with other members of the type-III copper protein family (see Table 2 for full names, PDB IDs, references and structural features). The *vs*Tyr core domain structure in cartoon representation is colored black, and the copper ions and bridging water molecule are shown as spheres in orange and red, respectively (in all five panels). For clarity, the copper centers of the homologous enzymes are not shown. The “arm-like” feature unique to *vs*Tyr consists of the three helices α4–α6 shown on its most left periphery. (**A**) *vs*Tyr superimposed with other bacterial tyrosinases that do not (**A**) or do (**B**) employ a caddy protein: (**A**) *bt*Tyr (light blue), *bm*Tyr (blue) (**B**) *sa*Tyr and its caddie protein (cyan and light pink), *sc*Tyr and its caddie protein (dark blue and light magenta). (**C**) *vs*Tyr superimposed with the closest plant-derived homolog *cg*AUS (green), *sl*Tyr (lime green), *ib*CO (forest green). (**D**) *vs*Tyr superimposed with two representatives of fungal tyrosinases and tyrosinase-like proteins: *tt*TyrP (orange), *abb*Tyr (pale yellow). (**E**) *vs*Tyr superimposed with the human tyrosinase-related protein 1. Further superpositions are shown in Appendix A.

**Figure 5 biomolecules-13-01360-f005:**
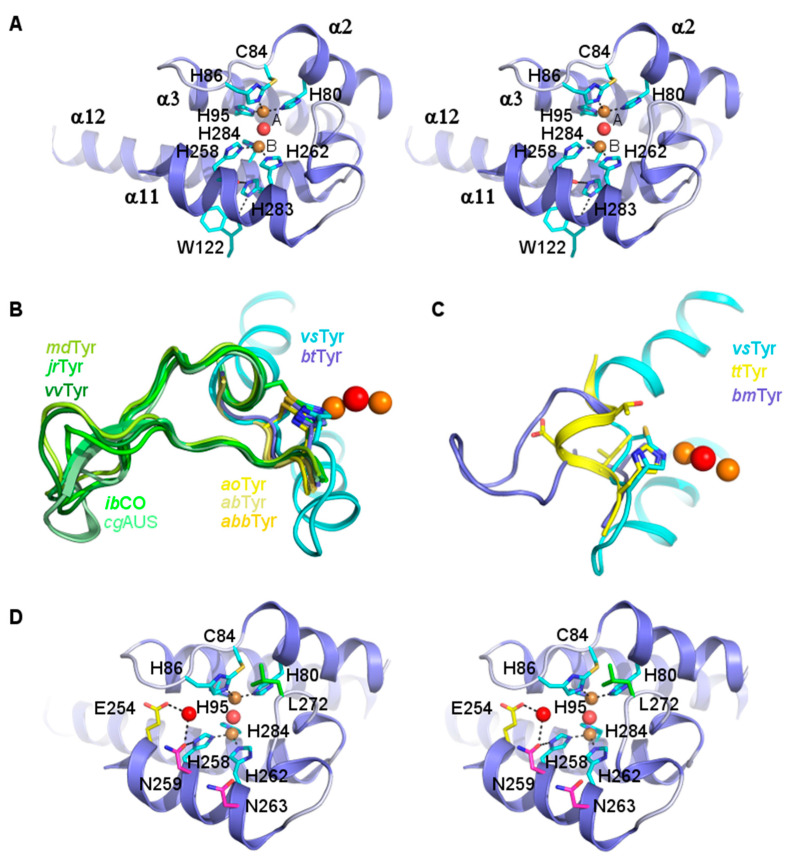
Structure of *vs*Tyr active site. (**A**) Stereo view of the binuclear copper center. The 4-helix bundle surrounding the metal center is shown as a cartoon, the copper-coordinating residues as sticks. CuA and CuB (orange spheres) are labeled A and B, respectively, with the bridging water molecule (red sphere) in between. The metal-coordinative and hydrogen bonds are dashed black lines. In the structure of *vs*Tyr, a seventh conserved histidine, H283, is not ligating a copper ion but is hydrogen-bonded to the side chain of W122 and the carbonyl oxygen of H284. C84 and H86 are covalently linked via a thioether bond. (**B**) Comparison of the thioether link topology of *vs*Tyr with plant and fungal PPOs. Helices α2 and α3 of *vs*Tyr, and the connecting loop carrying C84 and H86, are shown in cyan, the side chains of C84 and H86 as sticks with carbon atoms in cyan. Superimposed on to them are the corresponding thioether-linked residues and the α2–α3 connecting loop of the bacterial tyrosinase *bt*Tyr, the fungal tyrosinase *ab*Tyr, and of walnut tyrosinase (*jr*Tyr). For the plant enzymes *ia*CO, *cg*AUS, *vv*Tyr, and *md*Tyr, in which the thioether-linked residues adopt the same position and topology as in *jr*Tyr, and the fungal tyrosinases *ao*Tyr and* abb*Tyr, in which the link topology corresponds to that of *vs*Tyr, only the helix-connecting loop is shown. The color-coding is explained in the figure, the *vs*Tyr copper center is depicted as in A. (**C**) In *tt*Tyr (fungal, yellow) and *bm*Tyr (bacterial, slate blue), the second CuA-coordinating histidine (H118 and H60, respectively) is not covalently linked to a cysteine. Their superimposition with *vs*Tyr (cyan) H86 reveals that this is due to the absence of cysteine residues at a position corresponding to *vs*Tyr C84 in the helix-connecting loop, which also adopts a different conformation. (**D**) Stereo view of the *vs*Tyr active site. In addition to the dimetal-center coordinating residues (depicted as in A with carbon atoms in cyan), the figure also shows the position of the waterkeeper E254 (carbon atoms in yellow), the activity controllers N259 and N263 (carbon atoms in magenta), and that of L272 (carbon atoms in green). The latter corresponds to the position of the gatekeeper residue as seen in plant PPOs (see also Appendix A).

**Figure 6 biomolecules-13-01360-f006:**
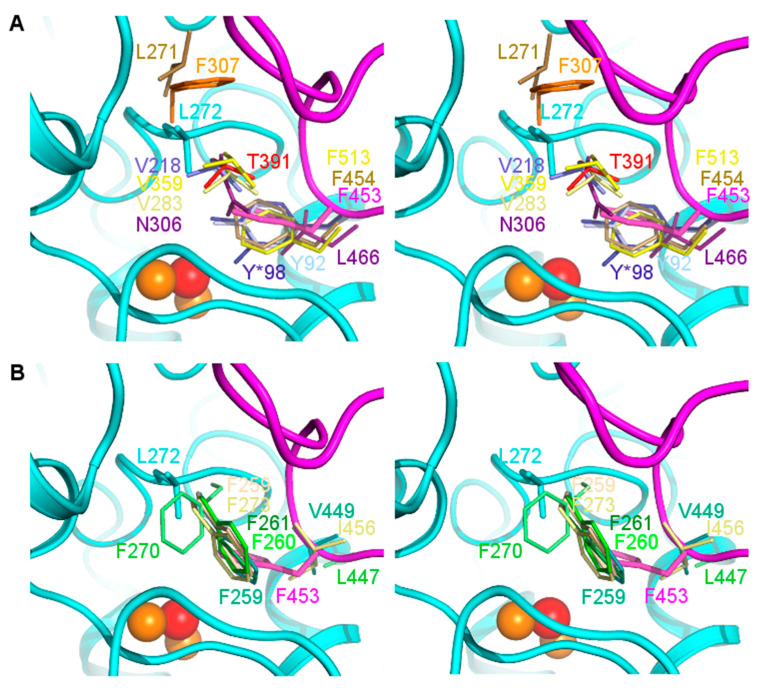
Gatekeeper residues in *vs*Tyr and homologous PPOs. (**A**) Stereo view of the gatekeeper positions in bacterial and fungal tyrosinases: *bt*Tyr (deep purple), *bm*Tyr (slate blue), *sa*Tyr (light blue), *sc*Tyr (dark blue), *ab*Tyr (pale yellow), *ao*Tyr (yellow), *tt*TyrP (orange), *abb*Tyr (light brown), superimposed with the corresponding position in *vs*Tyr core domain (cyan, L272) and C-terminal extension (magenta, F453, as modelled by Alphafold). *vs*Tyr, *sc*Tyr, *sa*Tyr, *ao*Tyr and *abb*Tyr contain an aromatic residue derived from the C-terminal extension or caddie protein whose position overlaps with that of the aromatic ring found at the original gatekeeper position in plant tyrosinases, whereas in *bt*Tyr this position is occupied by a non-aromatic side chain (L466). The precise placement of the amino acid side chains that, in the sequence, correspond to the original gatekeeper, i.e., to the aromatic side chain in plant tyrosinases, is much more variable. (**B**) Stereo view of the *vs*Tyr gatekeeper positions (colored as in A) compared to plant tyrosinases: *cg*AUS (pale yellow), slTyr (lime green), *ib*CO (dark green), *vv*Tyr (beige), *jr*Tyr (green), *md*Tyr (teal). The position of the alternative gatekeeper is unoccupied in the deposited structures of *ib*CO, *sl*Tyr, *jr*Tyr, *ab*Tyr, *bm*Tyr, and *tt*TyrP (as well as *hs*TyrRP1 not shown in the figure).

## Data Availability

The structure/structures have been deposited in the PDB with access codes: 8BBR (structure obtained after soaking with CuSO_4_), 8BBQ (structure obtained without soaking with CuSO_4_).

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
