# Peer review of "The Crystal Structure of Tyrosinase from Verrucomicrobium spinosum Reveals It to Be an Atypical Bacterial Tyrosinase"

_biomolecules, 2023, doi:10.3390/biom13091360_

Round 1

Reviewer 1 Report

The manuscript on the crystal structure of a tyrosinase from the bacterium Verrucomicrobium spinosum by Fekry et al. is straight forward and well written. This describes the atypical structure of this enzyme since it differs structurally in many aspects especially due to the absence of a caddie protein. The authors have cloned, expressed and crystallized a core domain of this enzyme and solved the structure at a reasonable resolution. The authors show that this core domain is like those of other type III copper enzymes although there is low sequence identity with the other homologous enzymes. In this aspect, the manuscript provides an incremental knowledge to the existing literature. The authors discuss the evolutionary aspects of the PVC superphylum which is interesting. To be able to understand the evolutionary aspect, more work needs to be undertaken but this manuscript adds another point to this knowledge. The authors point out that obtaining a co-crystal with substrates or inhibitors were unsuccessful with or without CuSO4; this is not surprising. It is not clear whether such co-crystallization experiments were performed using mutants of the gatekeeper residues. The authors also performed the modeling studies of the full-length molecule using AlphaFold.  Minimal data is presented for this part of the experiment. Comparison of the C-terminal domain structure with other homologous enzymes would help in understanding the diversity o this enzyme. Also, it is unclear whether biochemical assays were performed using the full length vs. core domain molecule. Although not required, it would be helpful in understanding the role of the C-terminal domain and possible from an evolutionary point of view.

Overall, this is a well written submission with very few modifications required.  

Adequate. Minor modifications needed.

Reviewer 2 Report

Fekry et al. have analyzed the crystal structure of the Tyrosinases of the bacterium Verrucomicrobium spinosum (vsTyr). Combined approaches, experimental (X-ray crystallography) and in silico (AlphaFold), to reveal the structure of vsTyr are used. The core domain structure is solved by X-ray crystallography and structures of other domains were inferred by AlphaFold prediction. Overall, the manuscript is well written, experiments are well designed, results are well represented and discussed. In my opinion, this manuscript will be of great interest for the readers working in the field of structural biology and enzymology. It can be acceptaed for publication after revision. I have following concerns about manuscript.  

#The recombinant core domain of vsTyr, expressed without the C-terminal extension, was shown to have a very high specific activity and to be unusually resistant to denaturing agents.

Authors state that the vsTyr has higher stability against denaturing agents but the structural attributes contributing to this stability haven’t been analyzed despite having nice high-resolution structure. This point should be included.

#The line Despite the low overall sequence identity of <30% with other type III copper enzymes. And Table 2.

This sentence needs to be reconsidered. Four out of sixteen compared structures as sequence identity higher than 30%. The value for “Query Coverage” should be provided in the “Sequence similarity column”.

#Figure 1

It is hard to visualize the secondary structure elements in 3D structure. I recommend that the representation of secondary structural elements on sequence map should be provided for better visualization. I am sure Author will find many in silico tools for that, but ESPript (https://espript.ibcp.fr/ESPript/ESPript/) is one among them can be used.  

#Figure 3 and 4

Figures 3 and 4 need to be shown with electron density map. It is important to represent that the model built goes well with the density map at the active site.

#Table 2

The I/σ(I) and CC1/2 values for high-resolution shell for 8BBR is suspiciously low. Should be cross-checked?

#Future direction

There are many reports/articles demonstrating that the tyrosinase is the potential candidate for rational bioengineering. It is disappointing that the presently solved structure of vsTyr is not analyzed and discussed for its capabilities to be bioengineered for industrial applications. This part should be added.  

Reviewer 3 Report

The paper by Fekry et al describes structural studies on an interesting tyrosinase from the bacterial species V. spinosum that shows some unusual features. The authors provide sufficient background about tyrosinases and why they are studying this protein. The experimental design is sound and the data are of sufficiently good quality. While the study largely features the crystal structure and does not include complementary functional studies or similar, the authors have performed some thorough structural analyses and present a number of interesting findings. They present an in-depth analysis of the crystal structure and comparison with that of structures of homologous tyrosinases from bacteria, plants, fungi and humans. The paper reads well and is easy to follow, the references are appropriate and the figures are high quality. I am particularly impressed that they use stereo-diagrams in figures, which is becoming rare these days.

Having said all this, it would improve the paper a lot if some further analyses were included. A number of analyses come to mind.

-       Considering the nature of the bacterium this protein comes from and the unusual features the authors uncovered, one example would be some bioinformatic and phylogenetic analyses on the evolution of these proteins

-       Another experiment would be to quantity the amount of copper ions per vsTyr molecule after copper reconstitution. In the results section 3.1, it was stated that crystals of vsTyr that were not soaked with 1 mM CuSO4 showed partially occupied copper sites. Inductive-coupled plasma mass spectrometry, ICP-MS could be used for this purpose. This would help to identify why there was no full occupancy at the two copper sites.  

-       Linked to the suggestion above, have there been any attempts to crystallise the metal-free vsTyr? What would be the structural changes in overall conformations and the active sites between the metal-free, and copper-bound vsTyr? Besides copper occupancy, what are the main structural differences between the unsoaked (8BBQ) and CuSO4-soaked (8BBR) crystals of vsTYR? This is not clearly stated in the main text.

-       Have the authors attempted to crystallize the pro-protein, rather than predict the structure by AlphaFold? See further comments on this topic below.

-       If the authors had made any mutants, they could also include them in the assay to further support the importance of the specific residues involved in enzymatic activity.

-       In Section 3.7, it is stated that the core domain of vsTyr has low affinity for the ligands and inhibitor. As such, attempts at co-crystallization and crystal soaking were unsuccessful. Have any binding affinity experiments (e.g. isothermal calorimetry, surface plasmon resonance, biolayer interferometry) been performed to support this statement?

The current manuscript without additional analyses is of average novelty and significance but would be of high interest to people involved in tyrosinase research. We leave the decision on whether any of the suggested analyses would be required for authors to respond to and at the discretion of the editors.

Below is a list of specific suggestions that may improve the paper further,  without performing further analyses and experiments.

-       Evolutionary origin of vsTyr: Lines 24-25 in the Abstract state “Our findings raise questions concerning the evolutionary origin”. This statement is rather uninformative – perhaps the work does not explain the evolutionary origin but some more informative statement would be useful in the abstract. The Conclusions section expands on the discussion of the evolutionary origin, but any conclusions are left rather vague. Some further computational work on the phylogeny would complement the structural analyses well and it is a pity it is not included here.

-       It would be good to explain why there was a need for pro-vsTyr to be expressed and then trypsinized to obtain the vsTyr core domain, instead of just relying on the vsTyr core domain construct for protein expression and purification. It would be good if you could provide a justification for it and the specificity of the trypsin digest step.   

-       What is the oligomeric state of the protein? Please mention this at the start of Results and Discussion somewhere and provide evidence for it.

-       Could the additional Cu ions found in one of the structures give some clues about the mechanism of metal loading into the high-affinity binding sites?

-       How stable is the purified pro-vsTyr? If crystallization attempts were unsuccessful, please state it in the main text.

-       In Section 2.2, the protocol for the vsTyr enzyme activity assay was described. However, the assay was only used to describe activity of the purified core domain of vsTyr and the purified pro-vsTyr. It would be recommended if the authors could present the results in a graph format instead of it being just a single sentence on lines 177-179. At the same time, they could also compare with that of the trypsinized pro-vsTyr to show that trypsin digestion had only cleaved away the c-terminal extension, leaving the core domain of vsTyr intact and giving it full enzymatic activity.

-       For Figure 1, it looks like the copper ions and water molecule are both in red. Please modify it. Ideally, it would be better to reduce the spherical size of the water molecule to differentiate it from that of the copper ions.

-       A multiple sequence alignment (MSA) of vsTyr for comparison with bacterial, plant, fungal and human tyrosinases would be beneficial and complementary to Table 2. This would help to highlight the conserved histidine residues involved in copper binding, the Cys-His thioether linkage and other residues involved in activity control, waterkeeping and gatekeeping. 

-       In Table 2, the plant protein slTyr has a “?” and “mutation?” for the Cys-His thioether linkage. Can you comment further what you meant by it?        

-       As AlphaFold2 was used to predict the full-length vsTyr structure, the authors should include the AlphaFold2 PAE plot in their supplementary materials and comment on the predicted model to that of their crystal structure of the core domain. They could also comment whether the overall structure of the predicted c-terminal extension is structurally similar to that of homologous structures from bacterial, plant and fungal. 

Minor corrections:

- Line 72: The full length vsTyr consists of 482 amino acids (residues 37-518), instead of 481.

- Line 115: “tobacco etch virus” does not need to be capitalized.

- Line 164: model building instead of modelbuilding

- Line 182: a protein doesn’t “crystallize in space group” – say something like : crystals have the symmetry of the space group”.

- Table 1: “freeb” – b should be superscript. Also please include a footnote with the software used to calculate the numbers.

- Fig. 3: yellow is hard to see.

- Line 275: I suggest “4-helix bundle”.

- Line 373: delete the comma.

- Line 406: something is wrong in this sentence, please rewrite.

- Line 462: “shows” (present tense)? It makes more sense when describing a structure – it still applies.

- References: something went wrong – there are no spaces.

- Line 504 and 505: CuSO4 and not Cu2SO4

See above

Reviewer 4 Report

Fekry et al. report the crystal structure of the catalytic domain of the enzyme tyrosinase from V. spinosum and discuss their finding in light of the structures of other tyrosinases and previous functional studies.

The article contains novel information and it is well written, and the structural analysis appears to be well done although the PDB report is missing.

The authors should do the following:

Provide de PDB validation reports.

Inform about the exact residues present in the purified polypeptide chain referring to the corresponding entry in the Uniprot data base.

Inform about the exact sequence of the N-terminal purification tag and which non-native residues remain after proteolytic cleavage.

Inform about the residues that could not be modeled (if any) due to insufficient electron density.

Explain if crystallization of the purified protein pro-vsTyr were ever tried (and the results). If it was not tried explain why.

Rewrite “3 600” removing the space or adding a comma.

Include another panel in figure S1 showing the predicted structure with the ribbon colored according to the colors used by AlphaFold to indicate the reliability of the prediction (pLDDT).

@font-face {font-family:Wingdings; panose-1:5 0 0 0 0 0 0 0 0 0; mso-font-charset:77; mso-generic-font-family:decorative; mso-font-pitch:variable; mso-font-signature:3 0 0 0 -2147483647 0;}@font-face {font-family:"Cambria Math"; panose-1:2 4 5 3 5 4 6 3 2 4; mso-font-charset:0; mso-generic-font-family:roman; mso-font-pitch:variable; mso-font-signature:-536870145 1107305727 0 0 415 0;}@font-face {font-family:Calibri; panose-1:2 15 5 2 2 2 4 3 2 4; mso-font-charset:0; mso-generic-font-family:swiss; mso-font-pitch:variable; mso-font-signature:-536859905 -1073732485 9 0 511 0;}@font-face {font-family:Times; panose-1:0 0 5 0 0 0 0 2 0 0; mso-font-charset:0; mso-generic-font-family:auto; mso-font-pitch:variable; mso-font-signature:-536870145 1342185562 0 0 415 0;}p.MsoNormal, li.MsoNormal, div.MsoNormal {mso-style-unhide:no; mso-style-qformat:yes; mso-style-parent:""; margin:0cm; mso-pagination:widow-orphan; font-size:12.0pt; font-family:"Times New Roman",serif; mso-fareast-font-family:"Times New Roman"; mso-ansi-language:EN-US; mso-fareast-language:FR;}p {mso-style-priority:99; mso-margin-top-alt:auto; margin-right:0cm; mso-margin-bottom-alt:auto; margin-left:0cm; mso-pagination:widow-orphan; font-size:10.0pt; font-family:Times; mso-fareast-font-family:Calibri; mso-bidi-font-family:"Times New Roman"; mso-ansi-language:EN-US; mso-fareast-language:EN-US;}.MsoChpDefault {mso-style-type:export-only; mso-default-props:yes; font-size:10.0pt; mso-ansi-font-size:10.0pt; mso-bidi-font-size:10.0pt; font-family:"Calibri",sans-serif; mso-ascii-font-family:Calibri; mso-fareast-font-family:Calibri; mso-hansi-font-family:Calibri; mso-bidi-font-family:Times; mso-font-kerning:0pt; mso-ligatures:none;}div.WordSection1 {page:WordSection1;}ol {margin-bottom:0cm;}ul {margin-bottom:0cm;}

Reviewer 5 Report

The article by Fekry et al. presents the crystallographic structure of the catalytic domain ( "core domain" (37-357)) of bacterial tyrosinase from V. spinosum. Complementing this structure, an alphaFold model of the whole enzyme is also provided to discuss the function of the C-terminal domain involved in the functional regulation of this enzyme. A priori, this study presents no novelty in the field of Tyrosinase enzymology, since there are almost twenty structures of this type of enzyme available in the PDB. The authors claim that the structure has shown its unexpected structural homology to plant tyrosinases, whereas sequence alignment, and in particular the identity % between sequences, already gives an idea of the similarities between these enzymes. I would advise the authors to submit this type of article to a specialized journal such as Acta Crystallographica Section F.

Some remarks:

- The paper would gain in clarity if a scheme showing the modular organization of vsTyr was presented with the delimitations of each domain

- Show the kinetic data from the activity tests mentioned in the paper

- It would also be useful to explain why the authors did not attempt to crystallize the full-length enzyme.

-The structural alignments in Figure 2 are extremely difficult to analyze, even with stereo glasses. As a key figure in the paper, this figure absolutely must be improved for greater clarity.

- Crystallization tests were carried out on the vsTyr catalytic domain in the presence of substrate ligands, but were unsuccessful. Were binding tests carried out beforehand to get an idea of the Kd values of the vsTyr domain for these ligands? 

- Docking of these substrates could be considered as an alternative.

- It would also be useful to show sequence alignments in the manuscript, even if this is in the supplementary materials. The sequence identity values shown in Table 2 are meaningless, as they are based on small portions of sequences. Sequence identity should be based on a complete alignment of all catalytic domains.

Round 2

Reviewer 2 Report

Again, very nice piece of work. Authors have provided satosfactory answers of most of my concenrs. The manuscript can be accepted after minor revision.

##The value for query coverage is already given in the table, in column 4 (labelled %Seq cov). We trust that the readers will understand that the “overall sequence identity” (i.e. for the full sequence) mentioned in the sentence the reviewer remarked on is not the same as the % seq id listed in the table that is observed for the part of the sequence identified as being homologous by BLAST.

Defination of "Overall sequence identity" should be provided in this case. 

#The I/σ(I) and CC1/2 values for high-resolution shell for 8BBR is suspiciously low. Should be crosschecked? Response: ##Yes, they are low. Traditional crystallography training would demand to cut the resolution at a level where I/σ(I) in the highest resolution shell is 2.0. However, nowadays high-resolution data beyond that is included in data analysis. The resolution cutting was performed by the autoprocessing at the synchrotron. We trust that their cutoff criteria are sensible and thus kept all the data.

Yes, I agree with that "2.0" is the traditional standard for I/σ(I). But the new standard also doesn't allow something less than 1.0. The I/σ(I) for the highest resolution shell of the said dataset is "0.6"!! (Not a big deal, but just a technical issue so wanted Authors to keep this in mind.

Reviewer 5 Report

The authors have drastically improved the manuscript. 

Enzyme activity results are now shown in the supplementary materials, but there are no standard devation values. These values must be entered in the table, together with the number of repetitions used to obtain them.

Round 3

Reviewer 5 Report

The authors have responded correctly to all the points raised.